# Modeling Complex Mathematical Reasoning via Large Language Model based MathAgent

## Abstract

Large language models (LLMs) face challenges in solving complex mathematical problems that require comprehensive capacities to parse the statements, associate domain knowledge, perform compound logical reasoning, and integrate the intermediate rationales. Tackling all these problems once could be arduous for LLMs, thus leading to confusion in generation. In this work, we explore the potential of enhancing LLMs with agents by meticulous decomposition and modeling of mathematical reasoning process. Specifically, we propose a formal description of the mathematical solving and extend LLMs with an agent-based zero-shot framework named **P**lanner-**R**easoner-**E**xecutor-**R**eflector (PRER). We further provide and implement two MathAgents that define the logical forms and inherent relations via a pool of actions in different grains and orientations: MathAgent-M adapts its actions to LLMs, while MathAgent-H aligns with humankind. Experiments on miniF2F and MATH have demonstrated the effectiveness of PRER and proposed MathAgents, achieving an increase of 12.3% (53.9% $\rightarrow$ 66.2%) on the MiniF2F, 9.2% (49.8% $\rightarrow$ 59.0%) on MATH, and 13.2% (23.2% $\rightarrow$ 35.4%) for level-5 problems of MATH against GPT-4. Further analytical results provide more insightful perspectives on exploiting the behaviors of LLMs as agents.

## 1 Introduction

Mathematical reasoning is a complicated task that requires the model to recognize the problem, associate domain knowledge, and determine the reasoning schema (Lu et al., 2023). Despite large language models (LLMs) (Ouyang et al., 2022; OpenAI, 2023) demonstrating interesting emergent abilities and impressive performances on NLP tasks (Wei et al., 2022a; Suzgun et al., 2023), solving complex mathematical problems can still be challenging (Borji, 2023). A possible reason is that LLM cannot accomplish all the required inferences for complex tasks in one step, leading to confusion and errors in recognition and generation. Recent studies attempt to extend the LLMs with agent-based systems to imitate the social interactive behaviors (Shinn et al., 2023; Park et al., 2023; Wang et al., 2023a), for instance, through task decomposition, cooperation, competition, and interaction with environments (Xi et al., 2023). Meanwhile, rudimentary concepts of decomposing the reasoning process into simple steps are also introduced to exploit the potential of LLMs, like CoT (Wei et al., 2022b; Wang et al., 2023c), Least-to-Most (Zhou et al., 2023b), and ToT (Yao et al., 2023). However, to the best of our knowledge, systematical decomposition and meticulous modeling of complex mathematical solving process have not been explored.

To tackle these problems, we summarize the characteristics of mathematical reasoning (MR): (1) The complex MR follows a **multi-step schema** (Yang et al., 2018), (2) The complex MR necessitates a fundamental capacity of **compound logical reasoning** (Allwood et al., 1982) for intermediate steps, and (3) **Domain knowledge** (Zheng et al., 2021) could be specifically requisite for advancing the inference in a single step. We conclude with a practical description of MR below. Fig. 1 illustrates a more detailed example. We can decompose the solving process into several steps that each aims to solve a sub-problem, performing conjunctive reasoning with an undescribed mathematical theorem: the "AM-GM Inequality".

**Description 1.** *Mathematical Reasoning (MR): Given a mathematical problem with conditions and a target, MR aims to implement a transition from the given conditions to the target through*

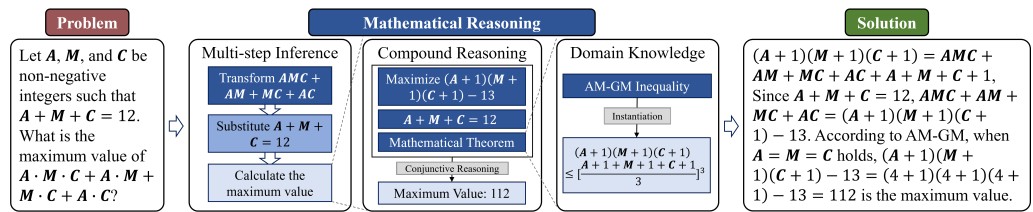

Figure 1: A case to illustrate the description of MR. Given a problem, MR includes three critical processes: decomposing the task, reasoning step by step, and introducing domain knowledge.

*finite-step $n$-fold inference. For each step, the inference process is executed by compound logical reasoning operations with corresponding mathematical knowledge.*

As the model size and training data volume increase, LLMs are already knowledgeable and adept at solving single-step problems (Bian et al., 2023). Therefore, when generalizing to complex mathematical problems, the key is to elicit LLMs parsing the underlying tasks, determining the reasoning schema, and dynamically recalling and associating relevant domain knowledge. Inspired by the concept of LLM-based agent systems, in this work, we first evolve a general mathematical agent framework, Planner-Reasoner-Executor-Reflector (PRER), to model the solving process of MR. PRER includes four critical modules. Planner and Reasoner are the main modules that perform step-by-step logical reasoning and filter the relevant knowledge to the problem. Executor accomplishes the current target by employing specific mathematical actions. Considering LLMs do not always perform correct calculations or make reasonable inferences, Reflector introduces a self-verification and self-correction mechanism to improve the stability and fault tolerance of the whole framework.

To deploy PRER in solving mathematical problems, we provide and implement two MathAgents in different grains and orientations, which mainly differ in Reasoner. MathAgent-M, the model-aligned system, defines actions that adapt to the inherent behaviors of LLMs. MathAgent-H, the human-aligned one, includes a human-aligned reasoner whose executable actions contain more expert-based prior knowledge. To evaluate the proposed MathAgents, we perform experiments on two complex mathematical benchmarks: miniF2F and MATH. Compared with GPT-4, MathAgents achieve improvements of 12.3% on the miniF2F dataset (53.9% → 66.2%), 9.2% on MATH (49.8% → 59.0%), and 13.2% (23.2% → 35.4%) for level-5 problems of MATH. Results against other baselines also demonstrate the effectiveness of PRER and two proposed MathAgents for MR. We then provide an ablation study of MathAgents and analyze the diverse behaviors of LLM agents. These analytical results further exploit the potential of LLMs for mathematical solvers.

In conclusion, our contributions include:

- To the best of our knowledge, we are the first to systematically decompose and model the solving process of complex mathematical reasoning, exploring the potential of integrating the LLMs with agents.

- We extend the LLM-based agent to mathematical reasoning by proposing a general zero-shot PRER framework. We also provide and implement two MathAgents based on PRER.

- Experiments demonstrated our proposed MathAgents outperform other baselines for complex MR, depicting the promising potential of PRER. Analytical results further reveal the detailed behaviors of LLM-based agents. We hope it inspires future research.

## 2 MODELING COMPLEX MATHEMATICAL REASONING VIA MATHAGENTS

### 2.1 FORMULATION OF MATHEMATICAL REASONING

Based on Description 1 and previous works (Lu et al., 2023), mathematical reasoning can be formulated by Equation 1, where $(X, y) \in D, X = \{x_1, x_2, \cdots, x_m\}$ represents the original conditions-target pair of a problem in the dataset $D$. For each step, conditions $X$ and the target $y$ are updated based on the domain knowledge $M$ and logical function $f_L$. When there exists a step $n$ such that $X_n \vdash y_n$ holds true, the reasoning is completed.

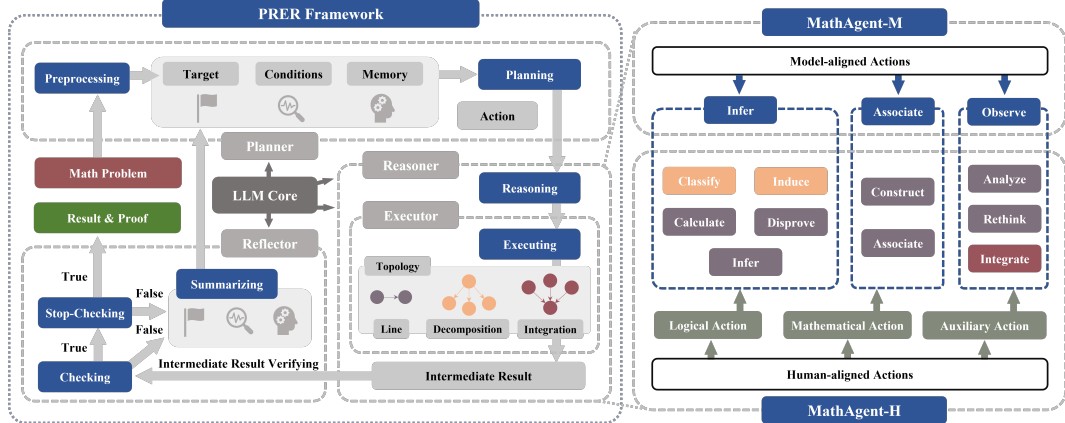

Figure 2: PRER framework with two MathAgent systems. The main difference between the two systems lies in the reasoner (and their corresponding executor).

$$\begin{aligned}
&\exists n \in \mathbb{N} : X_n \vdash y_n \\
&s.t.\ X_k, y_k = f_L(X_{k-1}, y_{k-1}, M), k = 1, 2, \cdots, n \\
&\quad X_0 = X, y_0 = y, (X, y) \in D
\end{aligned} \tag{1}$$

Except for some specific cases (like mathematical induction), $y$ is stable during the inference process as all proofs share the same target, the iteration can be simplified as $X_k = f_L(X_{k-1}, M)$.

As defined in the formulations that describe the multi-step process, the key is to design an open mathematical knowledge base $M$ and fit a general logical function $f_L(\cdot, M)$. Since LLMs have contained sufficient knowledge to simulate $M$, we introduce to extend LLMs with mathematical agents (MathAgent) to formulate $f_L(\cdot, M)$. Compared to previous works that introduced external tools to enhance the LLMs, we focus on constructing MathAgents purely by LLMs to explore the potential and behaviours of LLMs in building complex agent systems. More details of the evolution of agents are introduced in Appendix A.

## 2.2 PLANNER-REASONER-EXECUTOR-REFLECTOR

We propose a MathAgent framework, "Planner-Reasoner-Executor-Reflector" (PRER), to simulate the logical function $f_L(\cdot, M)$ shown in Figure 2. There are four critical modules of Planner, Reasoner, Executor, and Reflector with an LLM core.

**Planner** assumes the function of task decomposition (planning in the figure). For the $n^{th}$ step, when given the conditions $X_n$, the target $y_n$, and the memory $m_n$, the action $a_n$ can be selected by Equation 2, where $f_{PL}$ is the planning function in Planner.

$$a_n, m_n^1 = f_{PL}(X_n, y_n, m_n) \tag{2}$$

**Reasoner** adopts logical operations $f_{RS}(\cdot, M)$ to infer the intermediate result by Equation 3. In complex mathematical reasoning, some situations in Equation 1 cannot be tackled only by Reasoner. Therefore, we introduce **Executor** with three topologies to achieve the updation of $X_n$ and $y_n$.

$$X_n', y_n', t_n, m_n^2 = f_{RS}(a_n, X_n, y_n, m_n^1, M, f_{EX}(a_n, X_n, M)) \tag{3}$$

$f_{EX}$ is the executing function, and $t_n \in \{Line, Decomposition, Integration\}$ is the first-order topology of the inference. $Line$ is the linear topology that matches $X_k = f_L(X_{k-1}, M)$. $Decomposition$ and $Integration$ are adopted to fit Equation 1, and introduce non-linear structures and sub-tasks that can be solved by recursion.

Table 1: Action descriptions of both MathAgents. The Logical, Mathematical, and Auxiliary classes can roughly correspond to INFER, ASSOCIATE, and OBSERVE in MathAgent-M.

| Module | Class | Action | Description |
|---|---|---|---|
| Planner | - | Plan | Select the next-step action. |
| Reasoner (Agent-M) | - | Infer | Infer new rationales using deduction methods. |
| | - | Associate | Seek associations to uncover valuable external knowledge. |
| | - | Observe | Analyze and discuss existing conditions. |
| Reasoner (Agent-H) | Logical | Infer | Perform a general text-based reasoning. |
| | | Calculate | Focus on calculation or formula derivation. |
| | | Disprove | Involve negation or counterproof. |
| | | Classify | Perform classification discussion with finite cases. |
| | | Induce | Perform mathematical induction. |
| | Mathematical | Associate | Seek applicable theorems and formulas for the inference. |
| | | Construct | Design auxiliary conditions or variables. |
| | Auxiliary | Analyze | Guide the exploration via discussion or analysis. |
| | | Rethink | Think again outside the box. |
| | | Integrate | Integrate all sub-tasks. |
| Reflector | - | Check | Verify whether the inference is reasonable. |
| | - | Summarize | Summarize conditions/memories |
| | - | StopCheck | Determine whether reasoning needs to be stopped. |

**Reflector** is adopted to validate the effectiveness of the inference and to judge whether to stop or not by Equation 4, where $i_n$ is the stopping indicator of the inference process. Actually, Equation 4 contains three functions: validating, stop-checking, and summarizing. If the output of the validating function is "True", $(X_{n+1}, y_{n+1})$ is equivalent to $(X'_n, y'_n)$; Otherwise, $(X_{n+1}, y_{n+1}) = (X_n, y_n)$ holds. Stop-checking is to generate $i_n$, while the summarizing function is to summarize the updated conditions and target.

$$X_{n+1}, y_{n+1}, i_n, m_{n+1} = f_{RF}(X'_n, y'_n, X_n, y_n, m_n^2) \tag{4}$$

All these modules are simulated by the LLM core with diverse prompts. The final logical function $f_L(\cdot, M)$ is represented by the composition of $f_{PL}(\cdot)$, $f_{RS}(\cdot, M)$, $f_{EX}(\cdot, M)$, and $f_{RF}(\cdot)$, shown in Equation 5. In addition, the memory is always updated in all modules. Furthermore, Planner includes an addition function, preprocessing, to decompose the original problem into the form of $(X, y)$. This function does not participate in iterative reasoning.

$$f_L(\cdot, M) = f_{PL}(\cdot) \circ f_{RS}(\cdot, M, f_{EX}(\cdot, M)) \circ f_{RF}(\cdot) \tag{5}$$

Based on the PRER framework, we design two MathAgent systems: MathAgent-M and MathAgent-H, which will be illustrated in the next two sections.

## 2.3 MODEL-ALIGNED MATHAGENT AND HUMAN-ALIGNED MATHAGENT

Both our proposed MathAgents share a similar Planner, but differ at Reasoners and Executors. The Reasoner of the Model-aligned MathAgent (MathAgent-M) applies three coarse-grained actions: INFER, ASSOCIATE, and OBSERVE, as shown in Table 1. We do not impose any constraint on the reasoning process of MathAgent-M. Instead, the Planner are allowed to choose actions on its own and determine whether to execute the SUMMARY and STOPCHECK. Therefore, the solving process of MathAgent-M adopt a linear topology, which means we can directly update the conditions through Equation 6, where $f_M$ is the integration of $f_{RS}$ and $f_{EX}$ in MathAgent-M:

$$X_{n+1} = f_M(a_n, X_n, M), a_n \in \{\text{INFER}, \text{ASSOCIATE}, \text{OBSERVE}\} \tag{6}$$

We also allowed the Planner to make its own decision to invoke CHECK, but the results showed a certain decline. We therefore removed CHECK for MathAgent-M in all our experiments.

Human-aligned MathAgent (MathAgent-H) adopts a series of human-aligned actions, including logical , mathematical, and auxiliary actions (Table 1). Logical action imitates the process of human logical reasoning. For instance, INFER and CALCULATE introduce the similar conjunctive reasoning ($\wedge$) operator, while DISPROVE simulate a hypothetical reasoning ($\neg$) operator. INDUCE focuses on induction reasoning, incorporate prior domain knowledge, and CLASSIFY seeks to perform disjunctive reasoning ($\vee$). Logical actions can also be seen as refinements of INFER in MathAgent-M. Meanwhile, mathematical actions and auxiliary actions extend and refine ASSOCIATE and OBSERVE of MathAgent-M separately, providing more operations to complete the inference.

In addition, Executor introduces three topologies to imitate human thought (see Figure 2). For example, INFER and ASSOCIATE follow a $Line$ topology meeting the simplified equation, updating the conditions directly by combining newly generated and original conditions. An exception is DIS-PROVE, which needs to update $X_n$ and $y_n$ simultaneously, thus is the case of Formula Equation 1. CLASSIFY and INDUCE share the $Decomposition$ topology which transfers the original problem $(X, y)$ into a list of sub-problems $\{(X^i, y^i)|i = 1, 2, \cdots, m\}$. Instead, INTEGRATE re-integrates all sub-solutions. We can represented the updating by Equation 7, where $f_H$ is the integration of $f_{RS}$ and $f_{EX}$ in MathAgent-H.

$$
\begin{cases}
X_{n+1}, y_{n+1} = f_H(X_n, y_n, M), & a_n = \text{DISPROVE} \\
\{(X_n^i, y_n^i)|i = 1, 2, \cdots, m\} = f_H(X_n, y_n, M), & a_n \in \{\text{CLASSIFY}, \text{INDUCE}\} \\
X_{n+1} = f_H(\{(X_n^i, y_n^i)|i = 1, 2, \cdots, m\}, y_n, M), & a_n = \text{INTEGRATE} \\
X_{n+1} = X_n \cup f_H(X_n, M), & \text{Otherwise}
\end{cases}
\tag{7}
$$

**Execution on Mathematical Reasoning:**    To implement the proposed MathAgents for MR problems, we adopted a coupling method of LLM and programming. We prompt LLM to accomplish the main functions, and achieves logical connections by program. All prompts are depicted in Appendix B, and algorithms of MathAgent-M and MathAgent-H are presented in Appendix C.

## 3    EXPERIMENTS

**Datasets.**    We evaluate both our proposed MathAgents on two challenging datasets of mathematical reasoning: (1) MATH dataset(Hendrycks et al., 2021) which covers 7 topics with challenging competition mathematics problems, containing a total of 5000 questions for testing, and (2) MiniF2F dataset(Zheng et al., 2021) which comprises 488 high-school math competition questions, evenly divided into a testset and a valset. In our experiments, the Minif2f dataset provides final results or targets for proving, whereas the MATH dataset does not offer final answers for reasoning.

**Models.**    We conducted our experiments by utilizing the powerful GPT-4 ("gpt-4-0613") (OpenAI, 2023) as our fundamental model and baseline, employed via OpenAI API. To ensure the reproducibility, we adopt greedy decoding in all cases, i.e., the temperature is set to 0.

**Baselines.**    For the MiniF2F dataset, we compared our methods with (1) FMSCL (Polu et al., 2022) and (2) Autoformalization (Wu et al., 2022) that both are finetuned with expert iteration, (3) ReProver (Yang et al., 2023) that interacts with Lean by retrieval-augmented LLMs, (4) DSP (Jiang et al., 2022b) that prompting Minerva (Lewkowycz et al., 2022) to reason in a multi-stage schema, and (5) Decomposing-the-Enigma (Zhao et al., 2023) that decomposing the reasoning process into sub-goals, prompted with dynamic demonstrations selected by a specific diffusion model. For the MATH dataset, we compared our approachs with (6) WizardMath (Luo et al., 2023) and (7) MAM-MOTH (Yue et al., 2023) that both are specifically fine-tuned models, (8) CR (Zhang et al., 2023) that prompting LLMs to emulate human thoughts in a cumulative and iterative manner, (9) Complexity-Complexity-CoT (Fu et al., 2022) that provides a complexity-based selection strategy of demonstrations, and (10) PHP (Zheng Chuanyang, 2023) that improves the reasoning by hint-based iteration. Note that our proposed MathAgents are accomplished only by LLM cores under zero-shot setting.

### 3.1    MAIN RESULT

Our main results are shown in Table 2 and Table 3. We tested two proposed MathAgents and GPT4 based on "gpt-4-0613" with greedy decoding, and compared the baselines as reported in their papers.

Table 2: Accuracy on MATH dataset. We highlighted the best in bold, underlined the second best, and compare with GPT4 in the parentheses. CCoT denotes the complexity prompting (Fu et al., 2022). Besides, CR (Zhang et al., 2023) only used 500 test examples.

| Method | MATH Dataset | | | | | | | Overall |
|---|---|---|---|---|---|---|---|---|
| | Alg | Prob | Geo | InterAlg | NumTh | PreAlg | Precal | |
| *# previous reported SOTAs* | | | | | | | | |
| WizardMath (Luo et al., 2023) | 33.3 | 17.3 | 15.7 | 7.1 | 16.3 | 41.7 | 12.6 | 22.7 |
| MAmmoTH (Yue et al., 2023) | - | - | - | - | - | - | - | 46.8 |
| CR* (k=4) (Zhang et al., 2023) | 79.3 | 57.9 | 39.0 | 28.9 | 54.8 | 71.8 | 30.4 | 54.20 |
| GPT4+CCoT (k=8) (Zheng Chuanyang, 2023) | 70.8 | 53.1 | 36.5 | 23.4 | 49.6 | 71.6 | 26.7 | 50.36 |
| +PHP (k=8) (Zheng Chuanyang, 2023) | 74.3 | 56.3 | 41.9 | 26.3 | 55.7 | 73.8 | 29.8 | 53.90 |
| *# implemented based on GPT4 ("gpt-4-0613")* | | | | | | | | |
| GPT4 (OpenAI, 2023) | 66.3 | 53.5 | 41.5 | 23.7 | 43.0 | 74.5 | 29.7 | 49.76 |
| MathAgent-M (ours) | 64.3 | 54.6 | 44.1 | 27.2 | 45.4 | 74.4 | 31.5 | 50.88 |
| | (-2.0) | (+1.9) | (+2.6) | (+3.5) | (+2.4) | (-0.1) | (+1.8) | (+1.12) |
| Math Agent-H (ours) | **76.0** | **62.0** | **47.6** | **31.0** | **59.1** | **83.5** | **36.8** | **59.02** |
| | (+9.7) | (+8.5) | (+6.1) | (+7.3) | (+16.1) | (+9.0) | (+7.1) | (+9.26) |

Table 3: Accuracy on MiniF2F. We highlighted the best in bold, underlined the second best, and compare with GPT4 in the parentheses.

| Method | MiniF2F | | Overall |
|---|---|---|---|
| | valid | test | |
| FMSCL (Polu et al., 2022) | 33.6 | 29.6 | 31.6 |
| Auto (Wu et al., 2022) | 37.3 | 35.2 | 36.3 |
| ReProver (Yang et al., 2023) | - | 26.5 | - |
| DSP (Jiang et al., 2022b) | 43.9 | 37.7 | 40.8 |
| DeEnigma (Zhao et al., 2023) | 48 | 45.5 | 46.8 |
| GPT4 (OpenAI, 2023) | 56.6 | 51.2 | 53.9 |
| MathAgent-M | 62.3 | 61.5 | 61.9 |
| | (+5.7) | (+10.3) | (+8.0) |
| MathAgent-H | **65.6** | **66.8** | **66.2** |
| | (+9.0) | (+15.6) | (+12.3) |

Table 4: Ablation study on MiniF2F. We examine MathAgent-H by ablating a category of actions once.

| Method | MiniF2F-test | |
|---|---|---|
| | Agent-M | Agent-H |
| GPT4 | 50.0 | 50.0 |
| w/ agents | 56.0 | 71.0 |
| - Preprocessing | - | 62.0 |
| - Association | 53.0 | 52.0 |
| - Inference | 42.0 | 55.0 |
| - Observation | 51.0 | 47.0 |
| - Verification | - | 56.0 |
| - Summarization | - | 57.0 |

**MathAgents achieve SOTA performances on challenging mathematical datasets.** MathAgent-H exhibits an average enhancement of 9.26% over GPT4 on MATH, and outperforms baselines in all 7 topics. MathAgent-M demonstrates an average improvement of 1.12% compared to GPT4, and achieves the second best in 3 topics. On miniF2F, both agents surpass GPT4 and other SOTAs. In general, MathAgent-M achieves 61.9%, surpassing GPT4 by 8.0%, while MathAgent-H achieves a SOTA performance of 66.2%, showcasing an improvement of 12.3% over GPT4.

**Human-aligned MathAgent-H works better than model-aligned MathAgent-M.** Overall, MathAgent-H outperforms MathAgent-M in all topics, with a lead of 8.14% and 4.3% on MATH and MiniF2F respectively. An interesting phenomenon is that MathAgent-M has a decline compared to GPT4 in subjects of Algebra and PreAlgbra, which are relatively easier but requires more numerical calculations. We speculate that this is because the decomposition and the modeling of reasoning process magnify the inherent weakness in calculation of LLMs.

## 3.2 ABLATION STUDIES

We conducted ablation studies of MathAgents, using 100 test examples from MiniF2F, examining the preprocessing (dividing the statements and target), three action categories of Reasoner, and two components in Reflector. Results are displayed in the table 4, presenting overall decreases.

**Refined actions of Reasoner play a pivotal and unique role.** Results of MathAgent-M indicate that segregating and conceptualizing the reasoning process yield benefits. Interestingly, removing

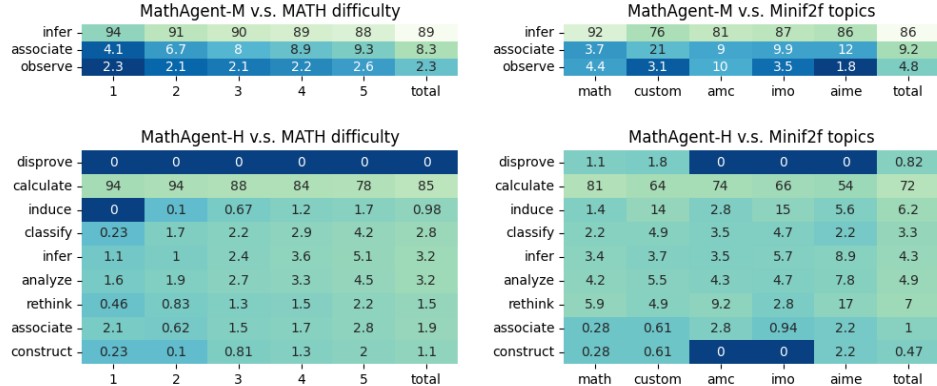

Figure 3: Frequency (%) of action scheduling at different difficulty levels/topics.

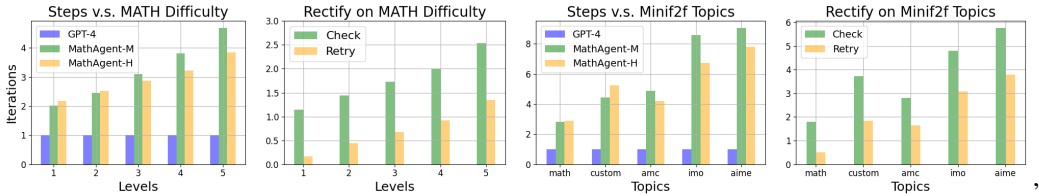

Figure 4: Steps of invoking actions and frequency of CHECK on different difficulty level for completing a reasoning. When CHECK determines an error occurs, the Reasoner will retry the last action.

such a coarse-grained action may cause LLMs' attempt to complete the delegated tasks via other actions. In contrast, removing modules of Reasoner resulted in larger decrease for MathAgent-H compared to Planner and Reflector. We hypothesize that the refined actions prioritize their own functions for further cooperation and thus are more essential and cannot be removed.

**Restricting and aligning LLMs by human thoughts can enhance the reasoning.** In MathAgent-M, we offer broad outlines of general actions for the LLMs to independently select their actions. Conversely, MathAgent-H establishes more rigid logical processes through fine-grained actions and programming, including preprocessing the problem and reflecting after each step. The results indicate emulating human thoughts to regulate LLMs could result in better reasoning.

### 3.3 ANALYTICAL RESULTS

We present analytical results to elucidate the behaviours of MathAgents. In Fig 3, we supply the statistical data of action calls. We further illustrate the required steps for accomplishing a problem in Fig. 4 and efficiencies of MathAgents at varying difficulty levels in Fig. 5.

**Refined actions can cooperate better to tackle complex tasks.** Fig 3 shows INFER and CALCULATE played the dominant role. When the difficulty increases, their occurrences tend to diminish. We observe that MathAgent-H exhibits a consistent shift in actions' proportion, where other actions may serve as collaborators for CALCULATE. Conversely, the proportion in MathAgent-M fluctuates slightly, suggesting the coarse-grained actions could be less effective for collaboration.

**MathAgent-H achieved better trade-off between inference and cost with CHECK.** As shown in Fig. 4, the average steps increase for both agents when the difficulty increases. The cost of MathAgent-H is relatively less, but it invests more in CHECK and retry, indicating that LLM struggles to complete complex tasks once but improves through past failures. Fig. 5 shows that both agents advance especially on complex tasks and CHECK leads to a high precision, i.e., correctly

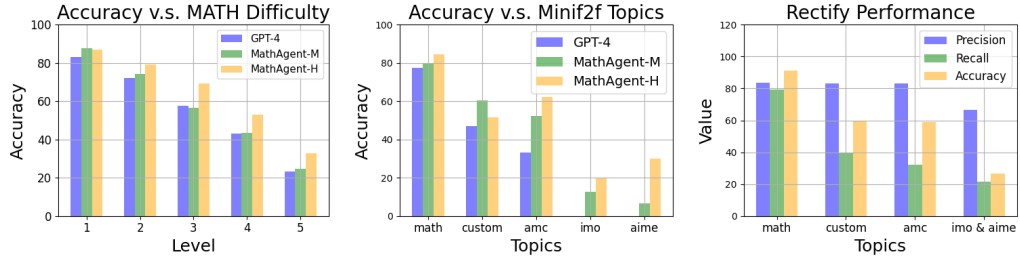

Figure 5: Accuracy on different difficulty levels of MATH and MiniF2F, as well as the performances of CHECK in MathAgent-H on 100 samples of MiniF2F-test.

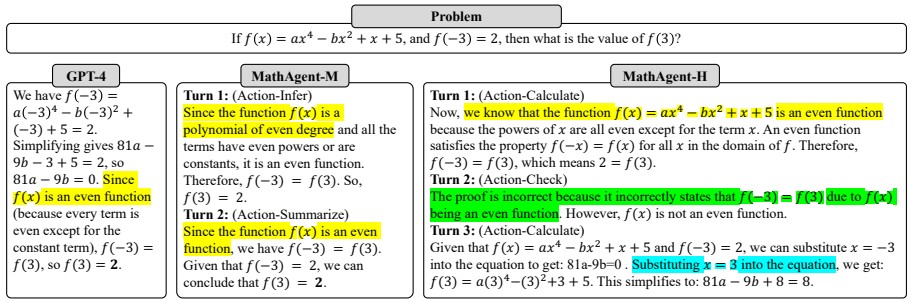

Figure 6: Case study on miniF2F. The highlights in yellow are incorrect propositions, while the highlighted green part indicates this mistake. The highlight in blue shows a revised proposition.

identified true answers. However, the recall rate could be relatively poor. We therefore suggest to retry last action to mitigate the harm when CHECK is wrong, instead of generating any guidance.

## 3.4 CASE STUDY AND ERROR ANALYSIS

We provide a case study to further understand the MathAgent systems. Figure 6 gives an example. GPT-4 mistakenly considers the function of $f(x) = ax^4 - bx^2 + x + 5$ as an even function. MathAgent-M also cannot revise this mistake. However, MathAgent-H finds this mistake and tackles the mathematical problem by CHECK. This case shows PRER's ability to revise mathematical hallucination of LLMs.

To evaluate the potential and constraints of PRER, we perform an error analysis as shown in Table 5, taking MathAgent-H as an example. There are four primary error categories: planning, execution, directional, and reasoning errors. We measure and analyze their percentages on miniF2F- test. Planning and execution errors are specific for MathAgents. We observe that the executing errors only account for $8.7\%$, while the proportion of planning errors is $25.9\%$, indicating the main limitation of MathAgents is that they may fail to

Table 5: Error analysis of PRER.

| ErrorType | Prop.(%) |
|---|---|
| P: select improper action. | 25.9 |
| E: systematic execution bugs. | 8.7 |
| D: lack of idea/knowledge. | 29.6 |
| R: calculation/symbolic error. | 35.8 |

parse the statements and devise a reasonable schema. Directional and reasoning errors also pose significant challenges ($29.6\%$ and $35.8\%$). For instance, the LLMs may not know the problem-solving strategies, the necessary formulas, or be prone to make computational errors and symbolical errors.

However, such directional and reasoning errors primarily stem from a lack of related knowledge or inadequacies in handling symbols and quantities. Although Reflector can alleviate the impact, it cannot fundamentally tackle these issues. Possible solutions include that pretraining LLMs with symbolic modeling internally, or introducing external tools as shown in Figure A1.

## 4 RELATED WORKS

### 4.1 ELICIT THE MATHEMATICAL ABILITIES OF LARGE LANGUAGE MODELS

Early works usually attempted to pre-train specific models for solving mathematical problems (Taylor et al., 2022; Lewkowycz et al., 2022; Zhao et al., 2022), based on carefully selected mathematical corpus. Since the reasoning capability of language model can significantly benefit from the general corpus like text and code (Wang & Komatsuzaki, 2021; Brown et al., 2020; Touvron et al., 2023, *inter alia*), some works also choose to fine-tune the general language model to elicit the informal mathematical abilities (Jiang et al., 2021; Han et al., 2021; Lample et al., 2022; Jiang et al., 2022a).

With the large-scale unsupervised pre-training technique, recent LLMs show interesting emergent abilities (Wei et al., 2022a), e.g., generalize from in-context prompts by learning the potential formats (Webson & Pavlick, 2022; Min et al., 2022; Pan et al., 2023). Instruction tuning further extend it to zero-shot by training LLMs to follow general instructions (Wei et al., 2021; Ouyang et al., 2022). One area of research therefore attempt to improve the performances by designing high-quality examples or instructing the model to generate executable programs of specific languages like Python code (Imani et al., 2023; Gao et al., 2022) or Isabelle/HOL code (Wu et al., 2022).

However, LLMs can struggle to directly generalize to complex problems and accomplish reasoning in a single step. Therefore, the idea of agent system has been introduced, trying to address this issue.

### 4.2 MODELING REASONING PROCESS VIA LLM-BASED AGENTS

One area of research that introduced the rudimentary concepts of agents is to prompt the LLMs to behaviour aligned with human, thereby implicitly decomposing the reasoning process (Wei et al., 2022b; Wang et al., 2023c; Zhou et al., 2023b; Yao et al., 2023). Intuitive strategies have also been introduced to improve the reasoning, including devising plans prior to reasoning (Wang et al., 2023b; Hao et al., 2023), self-reflecting the generations by LLMs (Zheng Chuanyang, 2023; Qian et al., 2023; Miao et al., 2023), and voting based on results for better consistency (Wei et al., 2022b; Suzgun et al., 2022).

Another area of research attempts to construct an agent system of multiple LLM cores and external helpers. Reflection-based methods allow the LLMs to identify potential issues according to feedbacks of multiple sources from external environment (Zheng Chuanyang, 2023; Qian et al., 2023; Wu et al., 2023; Miao et al., 2023). Du et al. (2023) proposed to improve the performance by multiagent debate. The Graph structure is incorporated into the schema of reasoning for modeling the reasoning process (He-Yueya et al., 2023; Chen et al., 2023a; McNichols et al., 2023; Cao, 2023; Zhang et al., 2023), typically requiring an abundance of relevant examples to help the model to understand the process. Some recent work also augment LLMs with external tools like search engine and code interpreter (Chen et al., 2023b; Imani et al., 2023; Jie & Lu, 2023; Zhou et al., 2023a).

Compared to these studies, we focus on constructing an systematical agent system, entirely using LLM cores, decomposing and modeling the complex reasoning processes from both perspectives of model behaviours and human cognition. With our more comprehensive and systematic framework, we achieve a better result.

## 5 CONCLUSION

In this paper, we explore the formulation of complex mathematical problems and propose a novel agent-based framework PRER to decompose and model the reasoning process. We further provide and implement two MathAgents, exploring the potential of intergrating LLMs with agent system. The results on challenging MATH ($59.0\%$) and miniF2F ($66.2\%$) demonstrate the effectiveness.

However, the proposed MathAgents still have constraints in calling expense, and could be harder to be implemented by weaker LLMs, including GPT-3.5. In our experiments, GPT-3.5 struggles with planning and comprehending the action, also exhibiting interesting phenomenon that provides repetitive analysis, but does not execute the actions. Moreover, the current prompts are manually crafted, heavily reliant on experts. We will tackle these issues in future research.

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

## A    APPENDIX: EVOLUTION OF MATHAGENT

Figure A1: (a) shows the BDI agent model. (b) is a general, practical LLM-based agent framework. (c) illustrates MathAgent.

LLM-based agent originates from agent-based model (ABM) following a theoretical agent model, Belief-Desire-Intention model (BDI). BDI defines an agent as "based on the **belief** about the state of the world and the **desire** for an ideal state, an **intention** to achieve the ideal state is formed, which generates the **action** that interacts with the external environment, accepts the **response** given by the environment, and further updates the belief". Based on the definition, Figure A1(a) shows a graphical illustration of BDI. Recent LLM-based agents can be counted as instantiations of BDI. Figure A1 shows a general, practical LLM-based agent framework, which defines a specific interaction method with the environment - Tool Using.

The most significant difference between MathAgent and a general LLM-based agent is that Math-Agent does not require an explicit environment to interact with but needs to complete reasoning within the agent, which is shown in Figure A1. Although MathAgent can also enhance the calculation abilities of the reasoner by calling tools, these external modules are not included in the agent. Therefore, we focus on the systematic exploration of MathAgent, and the use of external tools is no longer within the research scope.

## B    APPENDIX: PROMPTS IN TWO MATHAGENTS

We show all prompts in two MathAgents. As for MathAgent-M, we define Plan, Integrate, Summarize, and three kinds of actions, whose prompts are given in Table A1. Plan is a fundamental function to determine the present action with a JSON format. Integrate and Summarize are two auxiliary function to help the completion of the inference. Three actions, including INFER, ASSOCIATE, and OBSERVE, have been introduced in Section 3.2. All these actions are selected automatically with the help of Planner.

Table A1: Prompts in MathAgent-M

| Function | Prompt |
| --- | --- |
| Plan | Choose an action that could be helpful for solving the problem. The outputs should be in the JSON format of "{'Action': X, 'task': E}", where E is the objective or guidance of the action. |
| Infer | Infer new rationales within the given context using deduction methods such as equation transformation, calculation, induction, and more. The outputs should be in the JSON format of "{'inferences': Is}", where Is should be in one sentence. |

Table A1(continued): Prompts in MathAgent-M

| Function | Prompt |
|---|---|
| Associate | Seek associations within the given context to uncover valuable external knowledge. This knowledge may encompass theorems, lemmas, clever tricks, or any other insights not yet present in the context. The outputs should be in the JSON format of "{'associations': As}, where As should be in one sentence." |
| Observe | Analyze and discuss existing conditions, including generalization, negation, and reflective summarization. The outputs should be in the JSON format of "{'Observations': Obs}". |
| Summarize | Integrate intermediate steps into new premises and eliminate unnecessary or redundant references or rationales. The outputs should be in the JSON format of "{'new_premises': NPs}". |
| StopCheck | Summarize all generations into a final solution. The output should be in the JSON format of "{'status': ST, 'solution': Sol}". ST is either "solved" or "failed" and Sol is the summarized solution or the error analysis, specifically. |

MathAgent-H contains Plan, Check, Summarize, and three kinds of actions, whose prompts are given in Table A2. Plan is a fundamental function to determine the present action with a JSON format. Check and Summarize are two functions of Reflector. Logical actions (INFER, CALCULATE, DISPROVE, CLASSIFY, and INDUCE) is used to perform compound logical reasoning, while mathematical actions (ASSOCIATE, CONSTRUCT) is specific to mathematical problems. Auxiliary actions (ANALYZE, RETHINK, and INTEGRATE) help the system to complete the inference. All actions have been introduced in Section 3.3. In the plan function, {action: Action Description} is predefined, which is shown in Table A3, whose definitions are consistent with the result in Section 2.1.

Table A2: Prompts in MathAgent-H

| Function | Prompt |
|---|---|
| Plan | You are an AI planner specialized in choosing an action and designing the task/guidance. Please read the given context and choose an action that could be helpful for solving the problem. Then a special AI agent who is only capable of one action will be called to finish it by completing the task or following the guidance. {Question Description} Please choose one action based on following instructions: {action: Action Description} |
| Infer | You are an AI mathematician specialized in promoting the exploration and advance the reasoning/proving. Please read the given context and promote the proof. {Question Description} Remember, the only action you are capable of is defined as: {Action Description} |
| Calculate | You are an AI mathematician specialized in using calculation or formula derivation for promoting the exploration and advancing the reasoning/proving Please read the given context and promote the proof. {Question Description} Remember, the only action you are capable of is defined as: {Action Description} |
| Disprove | You are an AI planner specialized in solving mathematical problems by contradiction. Please read the given context and devise a contradiction scheme. {Question Description} Please design a contradiction scheme in the JSON list format of "{'Conditions': C, 'Goal': G}". C is all conditions assumed in proof by contradiction (including the necessary original conditions), and G is the target that is intended to be disproved in the proof by contradiction. |

Table A2(continued): Prompts in MathAgent-H

| Function | Prompt |
|---|---|
| Classify | You are an AI planner specialized in devising a classification discussion scheme to solve math problems. Please read the given context and devise a classification discussion scheme. Question Description Please divide the problem into some subproblems in the JSON list format of ”[{'Conditions': ST, 'Goal': SG}, and so forth.]”. ST is the new conditions of one subproblem and SG is the target of it. |
| Induce | You are an AI planner specialized in devising a scheme with mathematical induction method to solve math problems. Please read the given context and devise an induction scheme. Question Description Please divide the problem into two subproblems in the JSON list format of ”[{'Type': ”base step”, 'Conditions': C1, 'Goal': G1}, {'Type': ”induction step”, 'Conditions': C2, 'Goal': G2}]”. C1 and C2 are the new conditions of the base step and the induction step, respectively. G1 and G2 are the targets of the two steps. |
| Associate | You are an AI mathematician specialized in Seeking (external) applicable theorems and formulas to aid or start an exploration. Please read the given context and promote the proof. {Question Description} Remember, the only action you are capable of is defined as: {Action Description} |
| Construct | You are an AI mathematician specialized in constructing auxiliary conditions/variables to aid or start an exploration. Please read the given context and promote the proof. {Question Description} Remember, the only action you are capable of is defined as: {Action Description} |
| Analyze | You are an AI mathematician specialized in providing an analysis/discussion for further decision-making. Please read the given context and promote the proof. {Question Description} Remember, the only action you are capable of is defined as: {Action Description} |
| Rethink | You are an AI mathematician specialized in thinking outside the box or finding useful patterns for further decision-making. Please read the given context and promote the proof. {Question Description} Remember, the only action you are capable of is defined as: {Action Description} |
| Integrate | You are an AI mathematician who is good at summarizing the proof {Question Description} You need summarize the proof. Pay attention: the summary should be shorter and clearer in three sentences or less, and you don't need to judge the correctness of the proof. |
| Check | You are an AI mathematician who is good at checking proofs and summarizing them. Please read the given context and make your judgement. {Question Description} You need to check whether the proof processing is right. If you believe the proof is wrong, please output in the JSON format: ”{”Correctness”: ”wrong”, ”Summary”: R}”. R is the reason why you think the proof is wrong, and should be shorter and clearer in three sentences or less. Otherwise, if you think it's right, please output in the JSON format: ”{”Correctness”: ”right”}” |
| Summarize | You are an AI mathematician who is good at summarizing the proof Question Description You need to check whether the final target is solved. Please output in the JSON format: You need to check whether the final target is solved. Please output in the JSON format: ”{”Solved”:S1,”Summary”:S2}”. S1 is yes/no whether the final target is solved, S2 is your summary of this proof. Pay attention: S2 should be shorter and clearer in three sentences or less, and you don't need to judge the correctness of the proof. |

Table A2(continued): Prompts in MathAgent-H

| Function | Prompt |
|---|---|
| StopCheck | You are an AI mathematician who is good at summarizing the proof {Question Description} You need to check whether the final target is solved. Please output in the JSON format: You need to check whether the final target is solved. Please output in the JSON format: "{"Solved":S1,"Summary":S2}". S1 is yes/no whether the final target is solved, S2 is your summary of this proof. Pay attention: S2 should be shorter and clearer in three sentences or less, and you don't need to judge the correctness of the proof. |

Table A3: Action Description in MathAgent-H.

| Action | Description |
|---|---|
| disprove | 'disprove' involves negation or counterproof. |
| calculate | 'calculate' focus on calculation or formula derivation. |
| induce | 'induce' is using mathematical induction. |
| classify | 'classify' is using classification discussion with finite cases. |
| infer | 'infer' is general text-based reasoning when other actions are not applicable. |
| analyze | 'analyze' guides the exploration via discussion or analysis. |
| rethink | 'rethink' means think outside the box. |
| associate | 'associate' seeks applicable theorems and formulas. |
| construct | 'construct' designs auxiliary conditions or variables. |

## C  APPENDIX: ALGORITHMS OF MATHAGENT SYSTEMS

We also provide the execution algorithms of the two systems. MathAgent-M is a model-aligned system whose almost all actions can be selected by Planner automatically. Algorithm 1 shows the execution process of MathAgent-M.

---

**Algorithm 1:** MathAgent-M Algorithm.

**Input:** Problem $(X_0, y_0) \in D$, Simulation Function with LLM $f_{LLM}$, Prompts $P = \{$plan : $P_{\text{pl}}, \text{stopcheck} : P_{\text{sc}}, \text{summarize} : P_{\text{sm}}, \text{infer} : P_{\text{inf}}, \text{associate} : P_{\text{ass}}, \text{observe} : P_{\text{obs}}\}$
**Output:** Proof/Result $r$

1 Initialize Memory: $m_0 =$ NULL;
2 Initialize Stop Index: $idx =$ False;
3 Count: $n = 0$;
4 **while** $idx == False$ **do**
5      Select Action: $a_n, m = f_{LLM}(X_n, y_n, m_n, P_{\text{pl}})$,
       $a_n \in \{\text{infer}, \text{associate}, \text{observe}, \text{summarize}, \text{stopcheck}\}$;
6      Update Memory: $m_n \leftarrow m$;
7      **if** $a_n \in \{\text{infer}, \text{associate}, \text{observe}, \text{summarize}\}$ **then**
8          Execute Action: $X_{n+1} = f_{LLM}(X_n, P_A), A \in \{\text{inf}, \text{ass}, \text{obs}, \text{sm}\}$;
9      **else**
10          $idx, X_{n+1} = f_{LLM}(X_n, P_{\text{sc}})$;
11      **end**
12      **if** $idx ==True$ **then**
13          $r = X_{n+1}$
14      **end**
15      Count: $n \leftarrow n + 1$;
16 **end**
17 **return** r;

---

Based on Algorithm 1, MathAgent-M adopts a planning function to select actions in Reasoner and Reflector automatically. Afterward, the action is taken with a $Line$ topology. These two steps are executed alternately until the inference is completed or terminated.

MathAgent-H is a human-aligned system whose actions are defined to align with humankind. For example, INDUCE executes mathematical induction by dividing the problem into two sub-tasks: initial condition verification and induction, which obey demonstrated logic. Algorithm 2 shows the execution process of MathAgent-H.

In Algorithm 2, MathAgent-H also adopts a planning function to select actions in Reasoner automatically. Afterward, the action is performed by Reasoner with diverse topologies in Executor. Finally, Reflector checks the inference proof for each step. These three steps are executed alternately until the inference is completed or terminated. Note that not all actions are selected by Planner automatically. Instead, several specific actions, such as INTEGRATE, need to be performed at a fixed location, which is in line with human knowledge.

---

**Algorithm 2:** MathAgent-H Algorithm.

---

**Input:** Problem $(X_0, y_0) \in D$, Simulation Function with LLM $f_{LLM}$, Prompts
  $P = \{\text{preprocess} : P_{\text{pp}}, \text{plan} : P_{\text{pl}}, \text{summarize} : P_{\text{sm}}, \text{check} : P_{\text{ck}}, \text{stopcheck} : P_{\text{sc}}, \text{infer} :$
  $P_{\text{inf}}, \text{calculate} : P_{\text{cal}}, \text{disprove} : P_{\text{dis}}, \text{classify} : P_{\text{cls}}, \text{induce} : P_{\text{ind}}, \text{associate} :$
  $P_{\text{ass}}, \text{construct} : P_{\text{con}}, \text{analyze} : P_{\text{alz}}, \text{rethink} : P_{\text{rtk}}, \text{integrate} : P_{\text{int}}\}$

**Output:** Proof/Result $r$

1 Split Problem: $X_0, y_0 = f_{LLM}((X_0, y_0), P_{\text{pp}})$

2 Initialize Memory: $m_0 =$ NULL;

3 Initialize Stop Index: $idx =$ False;

4 Count: $n = 0$;

5 **while** $idx ==$ *False* **do**

6  Select Action: $a_n, m = f_{LLM}(X_n, y_n, m_n, P_{\text{pl}})$,
   $a_n \in \{\text{infer}, \text{calculate}, \text{disprove}, \text{classify}, \text{induce}, \text{associate}, \text{construct}, \text{analyze}, \text{rethink}\}$;

7  Update Memory: $m_n \leftarrow m$;

8  **if** $a_n ==$ disprove **then**

9   Prove by Contradiction: $X'_n, y'_n, m = f_{LLM}(X_n, y_n, m_n, P_{\text{dis}})$;

10   Update Memory: $m_n \leftarrow m$;

11  **else**

12   **if** $a_n \in \{\text{Classify}, \text{Induce}\}$ **then**

13    Classify: $\{(X_n^i, y_n^i)|i = 1, 2, \cdots, k\} = f_{LLM}(X_n, y_n, m_n, P_A), A \in \{\text{cls}, \text{ind}\}$;

14    Update Memory: $m_n \leftarrow m$;

15    **for** $i = 1$ to $k$ **do**

16     Recursive Calculation: $(X_n'^i, y_n'^i) = \text{Self}(X_n^i, y_n^i)$;

17    **end**

18    Integrate: $X'_n, y'_n, m = f_{LLM}(\{(X_n'^i, y_n'^i)|i = 1, 2, \cdots, k\}, y_n, m_n, P_{\text{int}})$;

19    Update Memory: $m_n \leftarrow m$;

20   **else**

21    Execute Other Action:
     $X'_n, m = X_n \cup f_{LLM}(X_n, m_n, P_A), y'_n = y_n, A \in \{\text{inf}, \text{cal}, \text{ass}, \text{con}, \text{alz}, \text{rtk}\}$;

22    Update Memory: $m_n \leftarrow m$;

23   **end**

24  **end**

25  Check: $idx_c, m = f_{LLM}(X'_n, y'_n, X_n, y_n, m_n, P_{\text{ck}})$;

26  **if** $idx_c ==$ *True* **then**

27   $X_{n+1}, y_{n+1} = f_{LLM}(X'_n, y'_n, P_{\text{sm}})$;

28  **else**

29   $X_{n+1}, y_{n+1} = X_n, y_n$;

30  **end**

31  Update Memory: $m_{n+1} \leftarrow m$;

32  Stop Check: $idx = f_{LLM}(X_{n+1}, y_{n+1}, P_{\text{sc}})$;

33  **if** $idx ==$ *True* **then**

34   $r = f_{LLM}(X_{n+1}, y_{n+1}, P_{\text{sm}})$;

35  **end**

36  Count: $n \leftarrow n + 1$;

37 **end**

38 **return** r;

---

