# OpenReview forum: "Modeling Complex Mathematical Reasoning via Large Language Model based MathAgent"
_ICLR.cc/2024/Conference — ICLR 2024 Conference Withdrawn Submission_

### Official Review · Reviewer_M3Nq · 2023-10-30

**Soundness:** 3 good
**Presentation:** 2 fair
**Contribution:** 3 good
**Rating:** 6
**Confidence:** 4

**Summary:**

This paper focuses on modeling the mathematical reasoning process within LLMs. The authors propose a novel framework named Planner-Reasoner-Executor-Reflector (PRER) and implement two MathAgents within this framework, i.e., MathAgent-M and MathAgent-H, to tackle complicated mathematical problems. The experimental results verify that the two agents significantly improve the reasoning accuracy.

**Strengths:**

1. The proposed PRER is a general framework whose idea of decomposing mathematical reasoning process is rational. Besides, the architecture of the paper is clear to read.
2. This framework can be implemented with different LLMs and in different grains. The motivation of describing LLMs’ and human-like behaviors is reasonable, and the corresponding technique makes sense.
3. The accuracy improvements on datasets MiniF2F and MATH are significant.

**Weaknesses:**

1. It’s necessary to give an \emph{overall} introduction of its idea. What I mean is not the description of the workflow of Planner-Reasoner-Executor-Reflector framework. What I am concerned about is the reason of formalizing the actions as “infer”、“calculate” and so on (Details could be referred to question 1 below). Besides, this paper presents several modules realized by prompting LLM, without a clear introduction to the internal logic and reasons for model design.
2. Some details lack enough descriptions or explanations, bringing difficulty in reproducing the proposed framework. Details could be referred to questions 2-4 below.
Please see the detailed questions below, which should be answered and addressed.

**Questions:**

1. How do the authors define the actions of different modules? For example, why does the “Mathematical” class in MathAgent-H only contain “associate” and “construct”? What is the behind idea of designing them? Are there other actions that should be considered?
2. According to the paper, the actions of MathAgent-M is a subset of MathAgent-H's. Therefore, what is the necessity of proposing MathAgent-M independently from the perspective of technique? Besides, as described in Table 1, the "Infer" action in MathAgent-M has different meaning with the "infer" in MathAgent-H. The authors state that the action in MathAgent-H is more aligned with human actions. However, the description of "infer" in MathAgent-M, i.e., "Infer new rationales using deduction methods" can also be viewed as an action in human cognition.
3. What is the meaning of m^1_n, m^2_n in Eqs.(2),(3) in section 2.2. They lack descriptions or explanations.
4. In Eq.(3), why is t_n (i.e., topology of the inference) an output, not an input, and even if t_n is obtained, what is it useful for subsequent reasoning? Because t_n does not appear in Eq.(4) or other equations.
5. According to Figure 3, “infer” and “calculate” occupy the most important part for MathAgent-M and MathAgent-H, respectively. It's a little weird for me due to the following reasons. First, as the authors have stated, MathAgent-H is more aligned with human actions. However, the statistics reveal that the human-like actions (e.g., "induce", "rethink") take up a very small proportion in MathAgent-H's reasoning process, contradictory with the motivation of designing MathAgent-H. Second, why "infer" takes up such a small proportion in MathAgent-H? Intuitively, since the testset are the same, "infer" should also be the prominent action in MathAgent-H, since it's more relevant to mathematical reasoning. On the contrary, "calculate" in MathAgent-H is indeed a computation action, which intuitively should not have such a high frequency.
6. There exist some other works that also decompose the mathematical reasoning into several steps (e.g., Tree of Thought [1]) and adopt a generate-then-verify paradigm (e.g., [2,3]). The authors need to give more illustrations of how this work is distinctive, explain the differences with other similar works, and incorporate them in experiments.
[1] Yao S, Yu D, Zhao J, et al. Tree of thoughts: Deliberate problem solving with large language models[J]. arXiv preprint arXiv:2305.10601, 2023.
[2] Charlie Chen, Sebastian Borgeaud, Geoffrey Irving, Jean-Baptiste Lespiau, Laurent Sifre, and John Jumper. Accelerating large language model decoding with speculative sampling. arXiv preprint arXiv:2302.01318, 2023.
[3] Yaniv Leviathan, Matan Kalman, and Yossi Matias. Fast inference from transformers via speculative decoding. In International Conference on Machine Learning, pages 19274–19286. PMLR, 2023.

---

> ### Author Response · Authors · 2023-11-15
> **Reply to Reviewer M3Nq**
>
> Thanks for your valuable comments and agreement with the concepts. We sincerely apologize for the lack of clarity in our writing. We summarized and replyed the frequent key issues among reviewers in "For All Reviewers" (*FAR*). Here are our specific replies:
>
> **C1: How do the authors define the actions of different modules?**
> R1: We supplemented the relevant theories in *FAR2* to explain how we define the modules and actions, including the introduction of logical reasoning and several empirical considerations.
>
> **C2: What is the necessity of proposing MathAgent-M independently?**
> R2: We explained the rationales for providing two MAs in *FAR3*, and clarified the motivation and contributions of this paper in *FAR1*. Although MA-H might be better, the comparision of two MAs can yield some interesting conclusions.
>
> **C3: Why the authors state the MA-M is more aligned with LLMs, while the description of "infer" in MA can also be viewed as an action in human cognition**
> R3: We primarily distinguish them by whether the LLM has the freedom to make decisions and self-evolve (see *FAR3*). The actions in MA-H is more specific and we contrain the LLM with program. Meanwhile, the actions in MA-M are more general. These actions like *observe* and *infer* are defined emperically, thus can contain human experiences. After all, language modeling based LLMs natually contains human knowledge. But overall, these actions allow the LLMs to execute freely.
>
> **C4: What is the meaning of $m^1_n$, $m^2_n$ in Eqs.(2),(3) in section 2.2. In Eq.(3), why is $t_n$ an output, not an input? What is it useful for subsequent reasoning?**
> R4: We re-organized the formulas in *FAR4*. For instance, when the LLM determines to *classify*, the LLM then need to seperate the context into sub-topics, that is $t_n$. This topology plays many roles in subsequent reasoning, as filtering memory to retain context only related to the sub-topic, and tell the LLM the relationships between sub-structures when finish discussion and integrat the memories.
>
> **C5: The *calculate* intuitively should not have such a high frequency.**
> R5: We elaborated what we observed about the actions in *FAR5*, analyzing why it is reasonable for *calculate* to have a higher proportion. However, we indeed observed some inconsistencies, which might make it abnormal. We provide further discussion in both *FAR5* and *FAR6*, explaining the possible reasons and solutions. It is also worth noting that as the difficulty increases, the proportion of other actions rises.
>
> **C6: There exist some other works that also decompose the mathematical reasoning into several steps.**
> R6: We provided some comparisons with existing work in *FAR1*, supplemented some unique theoretical foundations in *FAR2*, and analyzed the scalability of PRER in combination with these methods in *FAR6*. We will further improve the writing and supplement the required descriptions and comparisons in future revisions of this paper.

---

> ### Author Response · Authors · 2023-11-20
> **Looking forward to a discussion before the deadline**
>
> Dear reviewer M3Nq:
>
> Thanks very much for your attention and great effort in reviewing our paper. As the deadline is approaching and there are only less than **3 days** for discussion, we would like to have a further discussion with your feedback. We understand you may have a busy schedule, but we believe that we have addressed all your concerns. If you still have further concerns or feel unclear after reading our responses, please kindly let us know and we are willing to make clarification and discussion. If you are satisfied with our responses so far, we sincerely hope you could consider your score. Thanks very much!
>
> Best regards, Authors of #1727

---

> > ### Comment · Reviewer_M3Nq · 2023-11-23
> > **Reponse to the reviewers' rebuttal**
> >
> > Dear Authors,
> > Thank you for your detailed response to my comment. It is acknowledged that your responses have resolved some of my initial queries. I recommend revisiting the aspects as we discuss, ensuring that your work is presented in a manner that facilitates easy comprehension. Please consider incorporating more explicit explanations in the further version.
> > Best

---

### Official Review · Reviewer_CLmw · 2023-10-31

**Soundness:** 2 fair
**Presentation:** 2 fair
**Contribution:** 2 fair
**Rating:** 5
**Confidence:** 3

**Summary:**

The paper tackles improving the mathematical reasoning capability of LLMs by proposing a set of modular decomposition actions. These actions range from infer, associate, observe, disprove, etc. All the actions are simulated by LLMs with few shot prompts. With the help of these actions, the paper shows a strong performance improvement on MATH and MiniF2F datasets.

**Strengths:**

The paper attempts to systematically break down various useful actions in mathematical reasoning. The design is interesting.

The performance gain from prompting the LLM with proposed actions is quite significant especially on MiniF2F datasets, where it solves 20% IMO problems that are not solved before.

**Weaknesses:**

The paper is not very well-written with many of the equations not explained clearly. The authors should provide more clarifications on these.

The design of various actions seem heavily engineered and the overall algorithm quite complicated. (See Algorithm 2) I wonder if the authors could break down the effect of various actions and only identify a few that contribute to the performance improvement the most. This is especially important since according to Figure 3, majority of the actions are "calculate".

**Questions:**

1. What does this sentence mean: "whereas the MATH dataset does not offer final answers for reasoning"? I am very certain MATH datasets have ground truth reasoning steps and final answers.

2. What actions are truly necessary in improving the reasoning performance of LLM? Can the authors perform more thorough ablation?

3. Given the strong MiniF2F performance with MathAgent-H on IMO problems, can the authors provide a few generated proofs for those problems? Also, I was not able to find the prompts associated with MiniF2F.

4. Figure A1: where is (c)?

---

> ### Author Response · Authors · 2023-11-15
> **Reply to Reviewer CLmw (1/2)**
>
> Thanks for your valuable comments and suggestions about the soundness of this paper. We summarized and replyed the frequent key issues among reviewers in "For All Reviewers" (*FAR*). Here are our specific replies:
>
> **C1: More clarifications should be provided.**
> R1: We apologize for the deficiencies and omissions of writing. We supplemented the theoretical basis of MathAgents (MAs) in *FAR2*, and organized the key formulas of PRER in *FAR4*. We will further improve the writing.
>
> **C2: What actions are truly necessary?**
> R2: First it should be clarified that the core principle of our designing is the completeness of actions, rather than which one is important on MATH or MiniF2F (see *FAR2*). We provided more details of the action invocation in *FAR4*. For mathematical tasks, it is reasonable that the most important action is *infer/calculate*. Besides, we analyzed why the ratio of *calculate* is so high in *FAR4*, as well as the reasons and possible solutions. It's worth noting that when the difficulty increases, the frequencies of other actions also increase (Fig.3). We conducted the ablation experiments (Table 4). Considering the cost, more combinations of ablation might be hard.
>
> **C3: What does this sentence mean: "whereas the MATH dataset does not offer final answers for reasoning"**
> R3: We apologize for this error. The actual meaning is that when input to LLM, MATH does not provide the final answer, while MiniF2F informs the LLM of what the final goal to be proven is.
>
> **C4: Figure A1: where is (c)?**
> R4: Sorry for the typo error, the (c) here refers to the bottom "MathAgent" in the figure. We will correct the error later.
>
> **C5: Could the authors provide examples on IMO problems and more details of prompts on MiniF2F.**
> R5: An example is as follow:
> "Consider the system of equations
> $$\\left\\{\\begin{aligned}
> a_{11}x_1 +a_{12}x_2 + a_{13}x_3=0\\\\
> a_{21}x_1 +a_{22}x_2 + a_{23}x_3=0\\\\
> a_{31}x_1 +a_{32}x_2 + a_{33}x_3=0
> \\end{aligned}\\right.$$
> with unknowns $x_1,x_2,x_3$. The coefficients satisfy the conditions: (a) $a_{11},a_{22},a_{33}$ are positive numbers; (b) the remaining coefficients are negative numbers; (c) in each equation, the sum of the coefficients is positive. Prove that the given system has only the solution $x_1=x_2=x_3=0.$"
>
> Once encountering such equations, GPT4 immediately started equation transformations. However, this attempt is wrong and useless, only making it more and more complex. GPT4 will get confused by the long memory and start to make symbolic and logical errors. At the end, it might guess the answer, pretend it had successfully prove the problem, or just access the limitation of tokens.
>
> For instance, the generation of **GPT4** is:
> " (GPT4:) We can rewrite the system of equations as $[a_{11}x_1 = -a_{12}x_2 - a_{13}x_3,], ...$
> Then $[x_1 = \frac{-a_{12}}{a_{11}} x_2 - \frac{a_{13}}{a_{11}} x_3,], ...$
> Squaring each equation, we get..." (more equation transforms until reaching the token limitation)
>
> Relatively, based on the given actions, MA could standardize the behaviour of LLMs and making it possible to solve such complex problems. This does not mean the mathematical capacity of LLMs gets stronger, but the promtps change the solving direction.
>
> For instance, the **MA-M** first made associations and then infer correctly as:
> "(MA-m-associate:) The Rouché–Capelli theorem can be used to prove that a system of linear equations has only the trivial solution. This theorem states that...(detailed explanation).
> (MA-m-infer:) Since all the coefficients of the system of equations are non-zero, the rank of the coefficient matrix is 3. And since the right-hand side of the equations are all zeros, the rank of the augmented matrix is also 3....
> (MA-m-conclude:) According to the Rouché–Capelli theorem, this implies that the system of equations has a unique solution, which is the trivial solution."

---

> > ### Author Response · Authors · 2023-11-15
> > **Reply to Reviewer CLmw (2/2)**
> >
> > **MA-H** first detemined to prove the problem by contradiction, followed by a case-by-case discussion. For $x_1=0$, a correct contradiction was proven. For $x_1 \neq 0$, after two rounds of incorrect attempts, a correct contradiction was obtained. Finally, the conclusions were consolidated:
> > "(MA-h-disprove:) (some discussion)...Assume that there exists a non-zero solution for the system of equations,...(explain the goal:)Prove that this non-zero solution does not exist, which contradicts the assumption, thus proving that the given system has only the solution $x_1 = x_2 = x_3 = 0$.
> > (MA-h-classify:) class1: (some explanation)...Assume that $x_1 = 0$, $x_2$ and $x_3$ are not zero.
> > class2: (some explanation)...Assume that $x_1$, $x_2$ and $x_3$ are not zero.
> > ...
> > (MA-h-infer:) (for class1:) According to the assumption, if one of $x_2$ or $x_3$ has a value of 0, then according to the equation, the other must have a value of 0 for the $a_{11}x_1 + a_{12}x_2 + a_{13}x_3 = 0$ to be true. On the other hand, if neither of them is 0, it is known that $a_{12}$ and $a_{13}$ are negative, so there must be a positive number in $x_{2}$ and $x_{3}$, then this means that one of $a_{22}x_2 + a_{23}$ and $a_{32}x_2 + a_{33}x_3$ is positive and the other is negative.
> > ...(other discussion, check and integrate)
> > "
> >
> > Compared to GPT4, MAs decomposed the reasoning process and explored the search space more correctly.

---

> ### Author Response · Authors · 2023-11-20
> **Looking forward to your feedback before deadline**
>
> Dear reviewer CLmw:
>
> Thanks very much for your dedication in reviewing our paper. Your comments about some unclear details have been taken into account and prompted us to improve them in a revision of manuscript. Since there are only less than **3 days** to the deadline, we would appreciate it if you could check our replies and post further valuable comments about the replies and the presented example. We sincerely hope to continue the discussion with you and address the concerns.
>
> Best regards, Authors of #1727

---

> > ### Comment · Reviewer_CLmw · 2023-11-20
> > **Response to Authors**
> >
> > Dear authors,
> >
> > Thank you for the response. I have read the general comments and specific responses. It seems the authors also acknowledge the issues of cost and action importance. For cost, I believe authors could calculate the accuracy with respect of total tokens used and compare the baselines more fairly. For action importance, I appreciate the theoretical design choices authors make but I am also wondering how much performance would decrease if we only use the top-k actions exclusively. This would serve as a better ablation study.
> >
> > Another major issue is the MiniF2F evaluation. It seems that authors evaluate their method informally with natural language. This should usually be reasonable if a final answer is not provided and accuracy is calculated by checking if the output answer matches with the ground truth. However, for MiniF2F, the final goal to be proven is provided to the LLM. This makes evaluation not reliable since the model could concoct some invalid justification to prove the final goal. The baselines authors compared for MiniF2F are all formal theorem proving methods that do not suffer from this issue. Therefore, I believe the comparison on MiniF2F is not fair and reliable. Could the authors comment on this issue?
> >
> > Best regards,
> >
> > Reviewer CLmw

---

> > > ### Author Response · Authors · 2023-11-20
> > > **Reply to Reviewer CLmw**
> > >
> > > Dear reviewer CLmw:
> > >
> > > Thanks for your valuable comments and suggestions. We are pleased to engage in a more in-depth discussion with you.
> > >
> > > Regarding your first point, we appreciate your proposed approach for the ablation study. We will adopt it and refine our experiments. In addition, one key point we would like to emphasize is that as the model's overall reasoning ability is put to a more stringent test with the increasing complexity of the problem, relying solely on the model's ability to maintain correctness within just one or two steps is still very unreliable, even with a top-k voting approach. However, we believe it would be more meaningful to invest the same effort into building a reliable inference framework for a more effective and in-depth discussion of the problem.
> > >
> > > In response to your second point, we want to express our appreciation for your insightful question. When we chose the datasets, we noticed the same issue, so we selected Math (problems without results) and MiniF2F (problems with results), complementing each other in evaluating the PRER's capabilities.
> > >
> > > It is essential to clarify that the original form of the MiniF2F dataset is informal. Therefore, PRER's input is consistent with the original informal format of the MiniF2F dataset, and there is no unfair comparison with the baselines we chose.  The opposite truth is that some models perform an additional preprocessing step to convert the informal questions to a formal format, such as FMSCL, which means that PRER faces potential unfairness in the comparison. Furthermore, the distribution of difficulty levels in MiniF2F questions is relatively balanced, including representative questions spanning different difficulty levels. The number of questions is moderate, making it suitable for human manual review of the internal processes and testing the performance of PRER.
> > >
> > > However, it cannot be denied that the informal format of MiniF2F does provide convenience for models to solve the problems. Regarding this concern, the complete proof processes offered by the model on the MiniF2F were subjected to cross-validation by multiple individuals to ensure correctness.  We apply the same verification process to some reproducible baselines, such as GPT-4. On the other hand, we chose the Math dataset, which contained no answer information in the original questions, for a pure output accuracy test of PRER. Based on the experiments' results, our zero-shot framework still performs well on this dataset.
> > >
> > > Of course, in the future, we are also considering modifying the MiniF2F dataset, removing certain answer-related information, and conducting more in-depth experiments.
> > >
> > > Best regards, Authors of #1727

---

> > > > ### Comment · Reviewer_CLmw · 2023-11-20
> > > > **Response to Authors**
> > > >
> > > > Dear authors,
> > > >
> > > > Thank you for your response. I have a different understanding of MiniF2F and below are my thoughts.
> > > >
> > > > > It is essential to clarify that the original form of the MiniF2F dataset is informal.
> > > >
> > > > It is correct that the questions from the dataset were drawn from informal math competitions but the dataset itself focuses on formal theorem proving. The formal statements are in 4 different languages. Also, the title of the MiniF2F paper is "MiniF2F: a cross-system benchmark for formal Olympiad-level mathematics".
> > > >
> > > > > Therefore, PRER's input is consistent with the original informal format of the MiniF2F dataset, and there is no unfair comparison with the baselines we chose. The opposite truth is that some models perform an additional preprocessing step to convert the informal questions to a formal format, such as FMSCL, which means that PRER faces potential unfairness in the comparison.
> > > >
> > > > If the authors believe evaluating MiniF2F in the informal format is the right thing to do, can you explain why all baselines in Table 3 except GPT4 (devised by the authors) evaluate their method in the formal format?
> > > >
> > > > I would like to make a note that proving theorems in a formal environment is very challenging. What is typically omitted in the informal reasoning needs to be clearly stated and right premises need to be found to prove individual steps. If the authors truly believe PRER faces unfairness in the comparison, I would be happy to see and encourage authors to also run PRER in the formal format.
> > > >
> > > > Best regards,
> > > >
> > > > Reviewer CLmw

---

> > > > > ### Author Response · Authors · 2023-11-21
> > > > > **Reply to Reviewer CLmw**
> > > > >
> > > > > Dear reviewer CLmw:
> > > > >
> > > > > Thank you for your reply. As for your concerns and the points that may not be clear in the previous discussion, we will try to make a reply and supplement.
> > > > >
> > > > > In the results of miniF2F, we have carefully reviewed the evaluation methods of other approaches. No method relied entirely on informal language; they used prover to evaluate the correctness of formalized language proof processes. Our method, on the other hand, did not involve formalized language, and therefore, we did not use theorem provers in our evaluation.
> > > > >
> > > > > While the method of human evaluation might not be entirely objective, we rigorously assessed whether there were any missing steps in the intermediate process, whether the premises used in the proofs were complete, etc. Multiple checks by different people and discussion of doubtful problems are also adopted to reduce the deviation of manual evaluation.
> > > > >
> > > > > Our methodology (based on GPT-4) demonstrated a significant performance improvement over GPT-4 under the same evaluation conditions. We believe this sufficiently demonstrates the effectiveness of our framework. Of course, we will take the reviewer’s comments about the evaluation method seriously and aim to improve upon it in the future.
> > > > >
> > > > > Best regards, Authors of #1727

---

> > > > > > ### Comment · Reviewer_CLmw · 2023-11-22
> > > > > > **Response to Authors**
> > > > > >
> > > > > > Dear authors,
> > > > > >
> > > > > > Thank you for your response. Viewing the eagerness to tackle the addressed issues, I'm very confident that the authors can further polish the work and present an impactful work in their future submission.
> > > > > >
> > > > > > Best regards,
> > > > > >
> > > > > > Reviewer CLmw

---

### Official Review · Reviewer_WFXS · 2023-10-31

**Soundness:** 3 good
**Presentation:** 3 good
**Contribution:** 2 fair
**Rating:** 5
**Confidence:** 4

**Summary:**

The paper delves into the challenges LLMs face when solving intricate mathematical problems. To address these challenges, the authors introduce an agent-based zero-shot framework named Planner-Reasoner-Executor-Reflector (PRER) and two MathAgents, MathAgent-M and MathAgent-H. Experiments on miniF2F and MATH datasets show that the proposed approach significantly outperforms GPT-4, especially for level-5 problems of the MATH dataset.

**Strengths:**

1. The paper is well written and easy to follow.
2. The work pushed the state-of-the-art results on two datasets including MATH which seems to be a challenging dataset even for larger language models.
3. The implementation two MathAgents is innovative and shows promise in addressing the challenges LLMs face in mathematical reasoning.

**Weaknesses:**

1. Although the paper demonstrates advancements over GPT-4, it fails to specify the average model calls needed for each question within the PRER framework. This omission raises concerns about potential high costs. If the cost is, hypothetically, k times, would it still surpass k majority-voting? It would be advantageous to incorporate such an experiment. With succinct prompts, GPT-4 can outperform MathAgent-M on the MATH dataset. For instance, PHP achieves a score of 53.9.
2. The experiments are centered on particular datasets (miniF2F and MATH), both of which solely encompass abstract mathematical language. The efficacy of the proposed technique on other mathematical problem-solving datasets remains uncertain, especially concerning word problems akin to those in GSM8K. Such word problems can also be complex and may require more domain knowledge.
3. The paper's proposition of an approach that can “systematically decompose and model the solving process of complex mathematical reasoning” seems unsubstantiated with neither theoretical nor empirical backing. While using prompts to tailor LLM into a specialized expert is a prevalent strategy, the model doesn't acquire any fresh insights. Furthermore, there's an absence of empirical evidence emphasizing the significance or need for Executors.

**Questions:**

The paper introduces a technique to augment LLMs' aptitude in mathematical reasoning through an agent-based framework. Although the findings are encouraging, questions remain about the method's adaptability and the absence of exhaustive comparisons with alternative techniques.

**Correctness:** 3: Some of the paper’s claims have minor issues. A few statements are not well-supported, or require small changes to be made correct.

**Technical Novelty And Significance:** 3: The contributions are significant and novel, but there are areas that could be further explored or clarified.

**Empirical Novelty And Significance:** 3: The empirical contributions are significant, but the paper could benefit from a broader range of experiments and comparisons.

---

> ### Author Response · Authors · 2023-11-15
> **Reply to Reviewer WFXS**
>
> Thanks for your valuable comments and meaningful concerns about the costs, theories and soundness of this work. We summarized and replyed the frequent key issues among reviewers in "For All Reviewers" (*FAR*). Here are our specific replies:
>
> **C1: It raises concerns about potential high costs and trade-off.**
> R1: We explained and discussed the scalability and cost of PRER in *FAR6*, hoping these explanation can alleviate the concerns. Our main focus in this paper is to explore the potential of PRER, as stated in *FAR1*. We acknowledge the limitations and discussed how to reduce the costs as future research.
>
> **C2: Why not evaluating datasets like GSM8k.**
> R2: The framework we designed contains a variety of mathematical actions. Thus it may be meaningless to evaluate on GSM8k, which primarily requires computational operations, considering the cost. Previous methods have achieved very high acc on GSM8k, thus we chose more challenging datasets like MATH. Besides, MATH includes the topic of "algebra", which can reflect the generalizability of proposed MAs.
>
> **C3: It seems unsubstantiated with neither theoretical nor empirical backing. What's the significance or need for Executors.**
> R3: We apologize for the missing descriptions. We supplemented the theories in *FAR2* to explain how we define the actions and re-organized important formulas of PRER in *FAR4*. These contents will be added to the paper in the next version.
>
> **C4: How about the method's adaptability and comparisons with alternative techniques?**
> R4: We apologize for any confusion caused by our writing. We explained the scalability of PRER in *FAR6*. Theoretically, most prompting methods can be integrated as a type of agent. They are not alternatives, but rather compatible with PRER.

---

> ### Author Response · Authors · 2023-11-20
> **Looking forward to your feedback before the deadline**
>
> Dear reviewer WFXS:
>
> Thanks again for your constructive and valuable comments, which have helped us improve the paper considerably. We believe that your concerns have been addressed somewhat satisfactorily. There are only less than **3 days** to the deadline for discussions. Sincerely hope to gain further valuable feedback from you and continue to improve this work. We are also willing to clarify any additional concerns.
>
> Best regards, Authors of #1727

---

> > ### Comment · Area_Chair_jZAY · 2023-12-05
> >
> > Reviewer WFXS - please take a moment to read the final responses and decide if you would like to keep or change your rating. Thanks.

---

### Official Review · Reviewer_DbPQ · 2023-10-31

**Soundness:** 2 fair
**Presentation:** 2 fair
**Contribution:** 1 poor
**Rating:** 3
**Confidence:** 4

**Summary:**

In this work the authors develop a general agent-based framework, called Planner-Reasoner-Executor-Reflector (PRER), to model the problem solving process in mathematical reasoning (MR).
A feature of the proposed framework is that it only relies on LLMs, with no calls to external theorem provers.
The proposed approach is evaluated experimentally.

**Strengths:**

1) The experimental evaluation is rather thorough, the proposed approach is compared with several different frameworks.

2) The related literature is discussed in some detail, but it only mentions briefly related approaches that leverage on theorem-provers. Also, the paper does not discuss why avoiding the use of theorem-provers entirely.

**Weaknesses:**

1) The authors say that "to the best of our knowledge, systematical decomposition and meticulous
modeling of complex mathematical solving process have not been explored." However, there are a few pages on decomposition for mathematical reasoning, also cited by the authors themselves. Consider, for instance,

- Xueliang Zhao, Wenda Li, and Lingpeng Kong. Decomposing the enigma: Subgoal-based demonstration learning for formal theorem proving. arXiv preprint arXiv:2305.16366, 2023.

- Jason Wei, Xuezhi Wang, Dale Schuurmans, Maarten Bosma, Fei Xia, Ed Chi, Quoc V Le, Denny Zhou, et al. Chain-of-thought prompting elicits reasoning in large language models. Advances in Neural Information Processing Systems, 35:24824–24837, 2022.

- Shunyu Yao, Dian Yu, Jeffrey Zhao, Izhak Shafran, Thomas L Griffiths, Yuan Cao, and Karthik Narasimhan. Tree of thoughts: Deliberate problem solving with large language models. arXiv preprint arXiv:2305.10601, 2023.

and related:

- Tushar Khot, Harsh Trivedi, Matthew Finlayson, Yao Fu, Kyle Richardson, Peter Clark, and Ashish Sabharwal. Decomposed prompting: A modular approach for solving complex tasks, 2023.

2) Equation 1 is not entirely clear. It seems akin to the notion of deduction in logical systems, but its meaning is not formally specified. E.g., what is the meaning of symbol "|-"?

3) The different components of the proposed framework (planner, reasoner, executor, reflector) are presented rather in a hurry, in less than one page, by means of Equations (1) to (5), which are not explained in much detail either, especially as for the role of the different logical functions. Consider: "Planner
includes an addition function, preprocessing, to decompose the original problem into the form of (X, y)."
We don't get any more information about preprocessing in the paper.

4) As the authors themselves discuss limitations in the conclusions, "the current prompts are manually
crafted, heavily reliant on experts."
This might not be the most promising way forward.

**Questions:**

It is not entirely clear to me why two different agents, MathAgent-M and MathAgent-H, are required. What is the rationale for this choice?

---

> ### Author Response · Authors · 2023-11-15
> **Reply to Reviewer DbPQ**
>
> Thanks for your valuable comments and suggestions about the motivation and limitations of the work. We summarized and replyed the frequent key issues among reviewers in "For All Reviewers" (*FAR*). Here are our specific replies:
>
> **C1: Why avoiding the use of theorem-provers.**
> R1: We reasserted our contributions and design principles in *FAR1* and *FAR2*. Despite PRER can be combined with external tools, it is costly to provide additional implementation and the tool is not our contribution. Moreover, directly using theorem provers like lean wastes the textual ability of powerful LLMs. We will re-consider this trade-off in the future.
>
> **C2: There are a few pages on decomposition for mathematical reasoning.**
> R2: We addressed the comparison with previous works in *FAR1*. In short, we introduced more definitions and designs to tell the model how to decompose, and provided an agents-based framework with other concepts like logical reasoning to model the reason process.
>
> **C3: Equation 1 is not entirely clear. The framework is presented rather in a hurry.**
> R3: We apologize for the issues in writing. The formulas of the PRER are reorganized in *FAR4*. We will improve our writing and supplement required descriptions in latter paper revisions.
>
> **C4: The current prompts are manually crafted, might not be the most promising way forward.**
> R4: We acknowledge the limitations. Prompts have many advantages (interpretability, generalization, low cost, etc) and many drawbacks (see *FAR5* and *FAR6*). We plan to explore more methods like SFT in the future. Besides, recent work proposed some methods for automatic generation [1,2] and discrete search [3,4] of prompts, which may alleviate the concerns. Identify the intentions of non-experts to generate prompts is also an interesting direction to consider. The recent released "agents" by OpenAI imply the possibility.
>
> **C5: What is the rationale for choosing two MathAgents?**
> R5: We describe the how we define MAs in *FAR2* and explain why we provided two MAs and what's their diffence in *FAR3*.
>
> [1] Wang, Yizhong et al. “Self-Instruct: Aligning Language Models with Self-Generated Instructions.” Annual Meeting of the Association for Computational Linguistics (2022).
> [2] Honovich, Or et al. “Instruction Induction: From Few Examples to Natural Language Task Descriptions.” Annual Meeting of the Association for Computational Linguistics (2022).
> [3] Zhou, Yongchao et al. “Large Language Models Are Human-Level Prompt Engineers.” ArXiv abs/2211.01910 (2022): n. pag.
> [4] Sordoni, Alessandro et al. “Deep Language Networks: Joint Prompt Training of Stacked LLMs using Variational Inference.” ArXiv abs/2306.12509 (2023): n. pag.

---

> ### Author Response · Authors · 2023-11-20
> **Looking forward to a discussion before the deadline**
>
> Dear reviewer DbPQ:
>
> Thanks again for your great effort in reviewing our paper! Since there are only less than **3 days** to the deadline for discussions, we are really looking forward to having a discussion with you about all technical details and concerns. We sincerely hope to get your further feedback. Would you mind checking our response and letting us know if you have further questions?
>
> Best regards, Authors of #1727

---

> > ### Comment · Reviewer_DbPQ · 2023-11-22
> > **Follow up**
> >
> > Thanks to the authors for their comprehensive rebuttal and sorry for low reactivity.
> >
> > I think we all agree that the paper had serious flaws and it is not clear to me to what extent these flaws have been addressed in the revision.
> > I think the paper might benefit from a further round of proofreading. I'd suggest a resubmission to another venue.

---

### Author Response · Authors · 2023-11-15
**For All Reviewers: clarification on several key issues (1/2)**

We appreciate the insightful and valuable comments. To aid in the understanding of this paper, we provide several clarifications here and then respond to each reviewer separately. These clarifications will be incorporated in future paper revisions.

## FAR1: Clarification on our contributions and comparison with previous works.

We focus on decomposing and modeling the reasoning process of complex mathematical tasks, exploring the potential of proposed PRER and the behaviours of LLMs as agents. PRER shares similarities with many prompt and CoT-series methods, emphasizing the idea of "decomposing". However, previous works either directly ask the model to generate intermediate steps and sub-tasks/goals/questions (w/ or w/o examplers), or merely provide tree or graph structures to constrain the generation process. But they do not tell the LLMs **how to decompose** the reasoning process. Instead, PRER provides detailed actions to guide and constrain the behaviours of LLMs, furthermore mimicking some human cognitive processes as "memory" and "rectify". Prompting methods and external tools can be combined with PRER. Since they are not our contributions and the diverse implements can be costly, we focus on employing only LLM cores in this paper.

## FAR2: How do we define PRER, MathAgent (MA), and corresponding actions.

The proposed PRER is inspired by Belief-Desire-Intention (BDI)[1] (see Appendix A). We first employ PRER by the coarse-grained MA-M with 6 actions that empirically decompose the reasoning process as *infer*, *associate*, and *observe*.

In MA-H, we attempt to introduce the concepts of logical reasoning [2,3], including inductive/deductive reasoning [4,5]. Inductive reasoning in mathematics mainly refers to "mathematical induction" (*induce*), while deductive reasoning can be divided into simple propositions (*calculate*) and compound propositions. The compound proposition includes conjunction, disjunction, and hypothetical reasoning, respectively defined as *infer* ($\wedge$), *classify* ($\vee$), and *disprove* ($\neg$)[6].

We also empirically add some domain-specific actions, such as "theorem association" (*associate*) and "auxiliary variable/function construction" (*construct*), which guide LLMs to explore the search space and jump out suboptimal points. Finally, we introduce auxiliary actions such as *analyze*, *rethink*, and *integrate*, to assist the collaboration of the agents.

## FAR3: Differences between two MAs.

Both the design and implementation of the MAs are different. Some works allowed LLMs to make decisions and evolve autonomously [7], while others constrain the LLMs with empirical workflows [8]. Therefore we conducted two explorations: MA-M allows LLMs to accomplish the PRER freely with several general actions, while MA-H is constrained by specific actions and programs (see FAR2 and Fig.2). In addition, MA-H incorporated the concept of logical reasoning and adopted topological structures, including linear, decomposition, combination, etc. Therefore, we would state that MA-M aligns with LLMs and MA-H aligns with humans. As a result, the experiments present that MA-H is better.

[1] Andreas J. Language Models as Agent Models. EMNLP (Findings), 2022.
[2] Pan L, et al. A Survey of Deep Learning for Mathematical Reasoning. ACL, 2023.
[3] Barwise, J. An introduction to first-order logic. Studies in Logic and the Foundations of Mathematics, 1977.
[4] Martin E, et al. A General Theory of Deduction, Induction, and Learning. Discovery Science, 2021.
[5] Li Y, et al. MTR: A Dataset Fusing Inductive, Deductive, and Defeasible Reasoning. ACL (Findings), 2023
[6] Tafjord O, et al. ProofWriter: Generating Implications, Proofs, and Abductive Statements over Natural Language. ACL (Findings), 2021.
[7] Park J, et al. Generative Agents: Interactive Simulacra of Human Behavior. UIST, 2023
[8] Yao S, et al. Tree of Thoughts: Deliberate Problem Solving with Large Language Models. 2023

---

> ### Author Response · Authors · 2023-11-15
> **For All Reviewers: clarification on several key issues (2/2)**
>
> ## FAR4: The core formulas of proposed PRER and MAs.
>
> MA consists of four modules that are executed iteratively, solving the problem step by step. In each step, we infer the conditions, objectives, and memories {$x_{n+1}$, $y_{n+1}$, $m_{n+1}$} of step n+1 by inputing the corresponding {$x_n$, $y_n$, $m_n$}.
>
> Planner is to choose the current action $a_n$ by $$a_n = f_{PL}(x_n, y_n, m_n).$$ Then the Reasoner selects the current conditions $x_n'$, objectives $y_n'$, and topology $t_n$ to find proper context as the input of executor: $$x_n', y_n', t_n = f_{RS}(a_n, x_n, y_n, m_n, M),$$ where $M$ denote the domain knowledge provided via prompts. The executor performs the action to generate new memories: $$m_n' = f_{EX}(x_n', y_n', a_n, m_n, M)$$, and finally Reflector update the conditions, objectives, and memories via memories: $$x_{n+1}, y_{n+1}, i_n, m_{n+1} = f_{RF}(x_n', y_n', t_n, m_n')$$, where $i_n$ is used to determine whether the reasoning should be re-executing or early-stop.
>
>
> ## FAR5: More details and analysis of action execution/importances.
>
> The invocation/importance of actions is a common concern among reviewers. It should be first clarified that these actions are designed in theory, aiming to ensure MAs could tackle most mathematical problems, rather than designing specifically for MATH and MiniF2F. In experiments, the most important action is actually *infer/calculate*. Considering that MATH and MiniF2F are mostly multi-step calculations, a higher proportion of *infer/calculate* might be reasonable, also corroborated by SKiC [1]. It is notable that as the difficulty increases, the proportion of other actions gradually increases.
>
> In addition, there are indeed inconsistencies in the experiments as MAs are implemented via zero-shot prompts. We allowed powerful textual reasoning [2] of LLMs for *calculate*, thus expanding the meaning of such actions. Meanwhile, LLMs might not fully comply with the instructions and perform tasks outside the definition sometimes, as they are not pre-trained as collaborated agents. This interesting inconsistency led to a decrease in the invocation of other actions and made weaker models fail to understand and execute the actions. We will improve these issues by making agents more professional and independent in the future, using ICL, SFT, etc. In this paper, we focus on demonstrating the overall performances and potential and state the PRER and proposed MAs contribute.
>
> ## FAR6: Scalability and cost of proposed framework.
>
> The scalability of PRER is theoretically good. It can adopt different prompting methods (therefore they are compatible rather than alternative) as the strategies via instructions or workflow of agents via program. External tools and diverse feedback signals can further enhance LLMs.
>
> One primary concern about the provided MAs is that the presented instructions can only work well with GPT4. As analyzed in *FAR5*, the reason could be that weaker models fail to follow instructions since they were not pre-trained as agents. Methods like ICL, SFT, and combining external tools might be benificial to alleviate this concern.
>
> Another concern is the cost. Current MAs are expensive as calling GPT4 frequently. We reported the average steps in Fig.4. Similarly, many prompting methods chose to enhance LLMs by generating more times, e.g., PHP. The actual expense in our experiments is the memory (similar to problems encountered in RAG). Concurrent works like [3] have proposed effective method to manage memories, which might be a possible solution. Besides, we found outputs of LLMs could be repetitive and redundant and hallucination led to repeated execution of actions. We suggest these issues as future research.
>
> [1] Chen, J. et al. Skills-in-context prompting: Unlocking compositionality in large language models (2023).
> [2] Chen, Wenhu et al. Program of Thoughts Prompting: Disentangling Computation from Reasoning for Numerical Reasoning Tasks.” ArXiv (2022).
> [3] Chen, Howard et al. “Walking Down the Memory Maze: Beyond Context Limit through Interactive Reading.” ArXiv abs/2310.05029 (2023): n. pag.

---

### Comment · Area_Chair_jZAY · 2023-11-20
**Please engage in reviewer-author discussions**

Reviewers - I encourage you to read the authors' response carefully and let the authors know whether their response has addressed your comments.